# Divalent europium-doped near-infrared-emitting phosphor for light-emitting diodes

Jianwei Qiao[1,2], Guojun Zhou[1,2], Yayun Zhou[1], Qinyuan Zhang[1] & Zhiguo Xia[1,2]*

Near-infrared luminescent materials exhibit unique photophysical properties that make them crucial components in photonic, optoelectronic and biological applications. As broadband near infrared phosphors activated by transition metal elements are already widely reported, there is a challenge for next-generation materials discovery by introducing rare earth activators with $4f$-$5d$ transition. Here, we report an unprecedented phosphor $K_3LuSi_2O_7$:$Eu^{2+}$ that gives an emission band centered at 740 nm with a full-width at half maximum of 160 nm upon 460 nm blue light excitation. Combined structural and spectral characterizations reveal a selective site occupation of divalent europium in $LuO_6$ and $K2O_6$ polyhedrons with small coordination numbers, leading to the unexpected near infrared emission. The fabricated phosphor-converted light-emitting diodes have great potential as a non-visible light source. Our work provides the design principle of near infrared emission in divalent europium-doped inorganic solid-state materials and could inspire future studies to further explore near-infrared light-emitting diodes.

[1] State Key Laboratory of Luminescent Materials and Devices and Institute of Optical Communication Materials, South China University of Technology, 510641 Guangzhou, China. [2] The Beijing Municipal Key Laboratory of New Energy Materials and Technologies, School of Materials Sciences and Engineering, University of Science and Technology Beijing, 100083 Beijing, China. *email: xiazg@scut.edu.cn

Near-infrared (NIR) light source has attracted attention for applications in medical fields, bio-sensing, food processing industry, and night-vision technologies[1–3]. Traditional NIR light sources, such as halogen lamps and tungsten-halogen lamps, suffer from large sizes, low efficiency, short lifetime, and heat dissipation[4]. As an emerging field, researchers have made considerable efforts to explore new NIR emitters, such as InAs-ZnSe polymer nanocrystal, $CH_3NH_3PbI_{3−x}Cl_x$ organometal halide perovskite, $PbS@MAPbX_3$ (X = I, Br) quantum dots, and Pt(II) complex based organic light-emitting diodes (LED)[1,5–7]. However, the low efficiency, poor stability or complex fabrication technique restrict their development. As a contrast, the high luminous efficiency, long operating life, and compact sizes make NIR LEDs useful, however, a narrow full-width at half maximum (FWHM) below 50 nm seriously limit their applications in 3D sensing, food analyzing, and other specific fields[8]. Therefore, discovery of broadband-emitting NIR light sources is a necessary and urgent requirement. In this regard, phosphor-converted light-emitting diodes LEDs (pc-LEDs) have attracted increasing interests considering more opportunities for tunable emission during the phosphors' screening, and availability of simpler strategies for both materials design and fabrication of NIR pc-LEDs[9–12]. Furthermore, the low cost of NIR phosphors is beneficial. However, one of the most significant challenges is the discovery of blue light pumped and efficient broadband NIR-emitting phosphors.

Numerous studies on exploring NIR phosphors have been realized by doping transition metal ions ($Cr^{3+}$, $Ni^{2+}$, and $Mn^{4+}$), $Bi^{3+}$ or rare earth ions ($Pr^{3+}$, $Nd^{3+}$, $Tm^{3+}$, and $Yb^{3+}$) into inorganic host materials[13–19]. However, the sharp line emission of rare earth ions with 4f-4f transition is not broad enough for the desired applications, and the absorption region is also very narrow and weak. The $Ni^{2+}$ doped materials show broad emission band around NIR region, but the low efficiency and IR laser excitation suspends the application[20]. Recently, $Cr^{3+}$ ion has been considered as an ideal NIR luminescent center, which can exhibit broad-band emission ~650–1200 nm with $^4T_2 \rightarrow {}^4A_2$ transition[21–23]. However, mixed valence of $Cr^{6+}$ ions simultaneously exist in most $Cr^{3+}$ doped phosphors, which seriously quench NIR luminescent efficiency[21,24]. As one of important rare earth activators with 4f-5d transition, $Eu^{2+}$-activated phosphors normally possess high efficiency and have been applied in the field of commercial white LEDs. More importantly, the tunable emission in the visible range is very popular to discover useful phosphors, and the typical examples include $Sr[LiAl_3N_4]$:$Eu^{2+}$ (red), $\beta$-SiAlON:$Eu^{2+}$ (green) and $BaMgAl_{10}O_{17}$:$Eu^{2+}$ (blue)[25–27]. If $Eu^{2+}$-doped phosphors can realize NIR emissions, it would launch a new era in NIR light source. However, in addition to the recent report on $Ca_3Sc_2Si_3O_{12}$:$Eu^{2+}$, no other $Eu^{2+}$-doped phosphors can achieve NIR emissions with the peak beyond 700 nm[28]. Unfortunately, though $Ca_3Sc_2Si_3O_{12}$:$Eu^{2+}$ gives a broad-band emission ranging from 720 to 1100 nm, it cannot be pumped by blue LEDs due to the lack of blue light absorption of the sample[28]. Therefore, discovery of $Eu^{2+}$ activated NIR phosphors for blue light-pumped LEDs is an essential and difficult task for the emerging photonic, optoelectronic and biological applications.

The emission energy distribution of $Eu^{2+}$ is generally related with the average bond lengths, coordination numbers and the symmetry of the cations and corresponding polyhedron occupied by $Eu^{2+}$[29]. Many strategies, including neighboring-cation substitution, chemical unit cosubstitution and cation-size-mismatch have been used to tune visible photoluminescence of $Eu^{2+}$ ions[30]. However, there is still a challenge to realize the NIR emission for the difficulties of obtaining strong crystal field splitting (CFS) and large centroid shift originating from $Eu^{2+}$. Recently, our group has reported a new approach showing that $Eu^{2+}$ selectively occupies a site with small coordination number, which tends to

large redshift as found in $Rb_3YSi_2O_7$:$Eu^{2+}$ with red emission[31]. If the cations' sites occupied by $Eu^{2+}$ in $Rb_3YSi_2O_7$ are substituted by smaller ions, this effect will produce a larger CFS and even realize NIR emission. Accordingly, we have designed a $Eu^{2+}$-doped $K_3LuSi_2O_7$ NIR phosphor from isostructural $Rb_3YSi_2O_7$:$Eu^{2+}$ by substituting the large $Rb^+$ and $Y^{3+}$ ions with small $K^+$ and $Lu^{3+}$ ions to further increase the CFS effect[32].

Here in this work, we show that $K_3LuSi_2O_7$:$Eu^{2+}$ is the first $Eu^{2+}$-doped NIR phosphor under 460 nm blue light excitation and gives a broad-band emission peaking at 740 nm in the NIR region (600–900 nm). Structural and spectral analysis results indicate that the selective $Eu^{2+}$ site occupation in $LuO_6$ and $K2O_6$ polyhedrons contributes to the NIR light emission in $K_3LuSi_2O_7$:$Eu^{2+}$. The as-fabricated phosphor-converted LEDs have the potential to be used for non-visible light source, and the result could initiate the exploration of $Eu^{2+}$-doped NIR phosphors and other NIR spectroscopy applications.

## Results

**Crystal structure of $K_3LuSi_2O_7$:$Eu^{2+}$.** Figure 1a shows the simulated X-ray diffraction (XRD) profile of $K_3LuSi_2O_7$ and the measured XRD patterns for the $K_3LuSi_2O_7$:$xEu^{2+}$ (x = 0–0.05) with different $Eu^{2+}$-doping concentrations. All diffraction peaks can be well indexed by hexagonal cell ($P6_3/mmc$), and the parameters are similar to those of $K_3LuSi_2O_7$ phase, confirming the successful synthesis of $K_3LuSi_2O_7$:$Eu^{2+}$ phosphors[33]. To further understand the phase structure and sites occupation of $Eu^{2+}$ ions, Rietveld refinement were performed for $K_3LuSi_2O_7$:$xEu^{2+}$ (x = 0, 0.01, 0.02, and 0.04), as shown in Fig. 1b and Supplementary Fig. 1. All the refinements demonstrated low R-factors (Supplementary Table 1). Moreover, Supplementary Tables 2 and 3 listed the parameters of atoms' coordinates and bond lengths of the studied phases for the comparison.

$K_3LuSi_2O_7$ host (Fig. 1c) offers one Lu sites with 6-fold coordination, and two different K sites: K1 (9-fold) and K2 (6-fold), for the possible occupation by $Eu^{2+}$ ions. The cell volume increases with x (Fig. 1d), indicating that some $Eu^{2+}$ ions occupy Lu sites since the radius of $Lu^{3+}$ is smaller than that of $Eu^{2+}$ (Supplementary Table 4). However, the unit cell volume, V(x) shows a nonlinear increasing trend with increasing Eu content, thus another occupation mechanism should exist simultaneously, which competes with the main trend and reduces cell volume. Such mechanism can be ascribed to K → Eu replacement because K ion is bigger than Eu ion (Supplementary Table 4)[32]. To further explore the $Eu^{2+}$ selective site occupation, Fig. 1e illustrates the average bond lengths of $LuO_6$, $K1O_9$, and $K2O_6$ polyhedron in different concentration of $Eu^{2+}$ doped $K_3LuSi_2O_7$ by extracting the data from Rietveld refinement. With increasing $Eu^{2+}$ doping concentration, the average bond lengths of K1-O remain nearly unchanged, while the bond lengths of K2-O and Lu-O change significantly. In addition, the d(K2-O) and d(Lu-O) exhibit the same trend. It is owing to the fact that the occupation of Eu in Lu sites will lead to longer d(Lu-O) bond length, simultaneously, the amount of Eu in K2 sites is reduced and thus d(K2-O) bond lengths also become longer. Therefore, one can conclude that the main doping mechanism is ascribed to the synergetic effect of Lu → Eu and K → Eu replacements in $K_3LuSi_2O_7$, which is similar to the site occupation of $Eu^{2+}$ ions in isomorphic $Rb_3YSi_2O_7$[31]. The scanning electron microscopy (SEM) images and elemental mapping images in Supplementary Fig. 2 reveal that $K_3LuSi_2O_7$:$Eu^{2+}$ microcrystals are in well crystallized and K, Lu, Si, and O elements are homogeneously distributed.

**Photoluminescence properties of $K_3LuSi_2O_7$:$Eu^{2+}$.** Figure 2a gives the room temperature photoluminescence emission (PL)

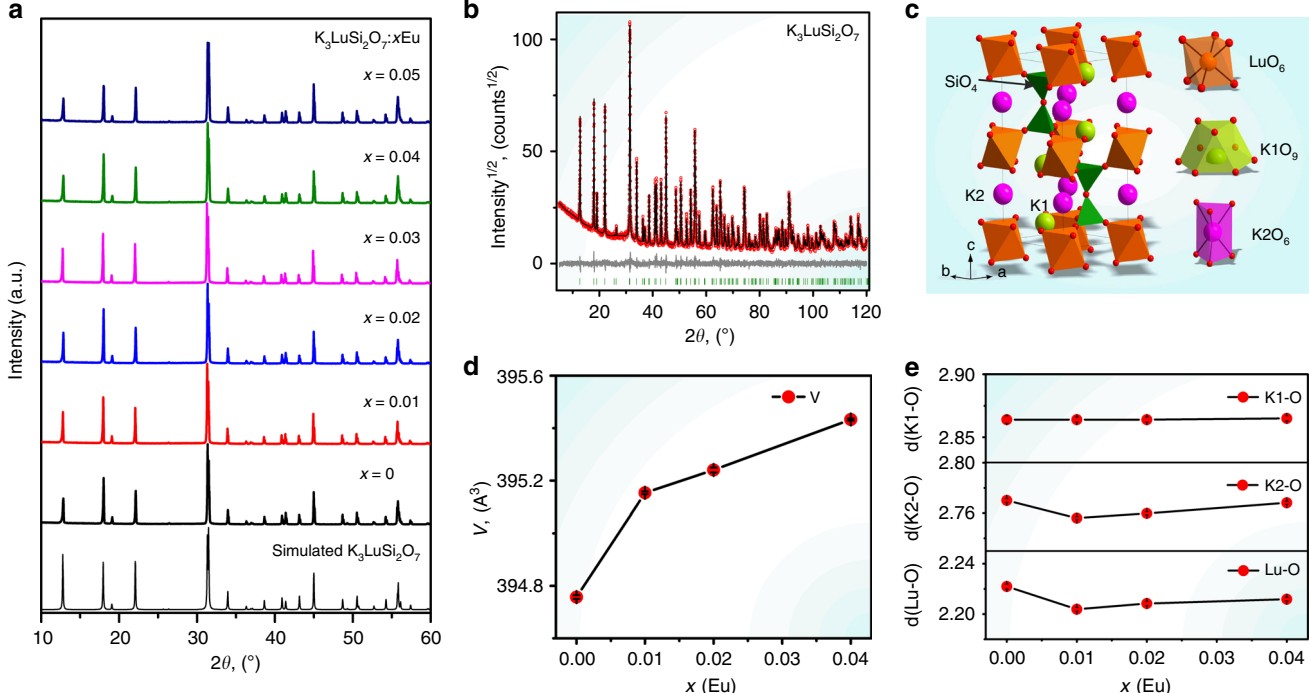

**Fig. 1** XRD patterns and crystal/local structure descriptions of $K_3LuSi_2O_7$:Eu. **a** Simulated and measured XRD patterns of $K_3LuSi_2O_7$ and $K_3LuSi_2O_7$:$xEu^{2+}$ ($x = 0.01$–$0.05$). **b** Rietveld refinement and **c** crystal structure of $K_3LuSi_2O_7$. **d** Cell volume $V(x)$ dependence of the Eu doping concentration ($x = 0$, 0.01, 0.02, 0.04). Small nonlinear trend can be associated with K → Eu replacement except for Lu → Eu substitution. **e** Dependence of the average bond length of Lu, K1, and K2 in $K_3LuSi_2O_7$:$xEu^{2+}$ ($x = 0$, 0.01, 0.02, 0.04)

and excitation (PLE) spectra of $K_3LuSi_2O_7$:$0.01Eu^{2+}$. When monitored at 740 nm, $K_3LuSi_2O_7$:$0.01Eu^{2+}$ gives an ultra-broad excitation band and a maximum excitation peak is found ~460 nm in the range of 250 and 600 nm. Such a broad excitation band should be ascribed to the 4 f → 5d transition of two different $Eu^{2+}$ sites in $K_3LuSi_2O_7$, as verified by the structural analysis discussed previously, and some sharp lines are related with the spectrophotometer itself. Supplementary Figure 3a illustrates the diffuse reflection spectra of $Eu^{2+}$ doped and undoped $K_3LuSi_2O_7$ samples. $K_3LuSi_2O_7$ host shows nearly no absorption from 300 to 800 nm, however, $K_3LuSi_2O_7$:$Eu^{2+}$ exhibits an ultra-broad absorption band from 250 to 600 nm with the introduction of $Eu^{2+}$ in the host. $K_3LuSi_2O_7$ exhibits a white body color under natural light suggesting a band gap of about 5 eV (Supplementary Fig. 3b). The diffuse reflection spectra of $K_3LuSi_2O_7$:$Eu^{2+}$ are in accordance with the excitation spectrum discussed above. Such a broad absorption band from ultraviolet to visible light region leads to deep red body color under natural light (the inset of Fig. 2a) and favors its application for blue light-pumped pc-LEDs. Thus, the NIR emission spectrum under 460 nm excitation reveals a broadband emission centered at 740 nm with FWHM ~160 nm (Fig. 2a). This emission is originated from the typical 5d → 4 f transition of $Eu^{2+}$. The asymmetric spectral profile demonstrates that there exists more than one luminescent center in $K_3LuSi_2O_7$:$Eu^{2+}$. To verify this point, the PLE/PL spectra depending on the different experimental conditions are measured in detail. With the increase of excitation wavelengths from 300 to 360 nm, the normalized emission spectra display a large redshift from 695 nm to 745 nm (Fig. 2b). But a small blueshift in emission is observed when the excitation wavelength is increased beyond 360 nm. Also, the peak at 315 nm in the normalized excitation spectra gradually decreased as the monitored wavelength increased from 660 to 800 nm (Supplementary Fig. 4a). These results proved that multiple luminescent centers existed in $K_3LuSi_2O_7$:$Eu^{2+}$. Moreover, under the excitation of the laser LED

light source with different excitation wavelengths at 375 nm and 450 nm, Supplementary Fig. 4b gives the normalized PL spectra of $K_3LuSi_2O_7$:$0.01Eu^{2+}$, suggesting the similar spectral profiles.

The broad emission band upon the excitation of 460 nm is exactly deconvoluted into two Gaussian spectral profiles with two peaks at 13900 $cm^{-1}$ (719 nm) and 12750 $cm^{-1}$ (784 nm), suggesting that $Eu^{2+}$ substitute two different cations in $K_3LuSi_2O_7$ (Fig. 2c). Figure 2d shows the PL decay curves of $K_3LuSi_2O_7$:$0.01Eu^{2+}$ measured at low temperature (80 K) and at room temperature (300 K) under 450 nm pulse laser diode excitation. The decay curves were comparatively fitted by mono- ($n = 1$), bi- ($n = 2$) and triple- ($n = 3$) exponential functions:[12]

$$I = \sum_{i=1}^{n} A_i \exp\left(-\frac{t}{\tau_i}\right), \quad (n = 1, 2, 3) \quad (1)$$

where $I$ denotes luminescence intensity, $t$ represents time; $\tau_i$ is lifetime for different components; and $A_i$ is the corresponding fitting constants. As compared in Fig. 2d, the decay curve measured at 80 K is well fitted by both the bi- and triple-exponential function, and two fitting curves nearly overlap. However, the decay curves at 300 K can be only fitted by the triple-exponential function, which is ascribed to an additional room temperature nonradiative transition owing to the lattice thermal vibration. Accordingly, the average lifetime values are 1.58 μs (at 300 K) and 2.28 μs (at 80 K), respectively. The detailed fitting results are presented in Supplementary Table 5. Moreover, the decay times are shortened with increasing Eu concentration (Supplementary Fig. 5a) due to the energy transfer between different $Eu^{2+}$ luminescent centers. Upon the excitation of 340 nm pulse laser diode, the average lifetime (1.69 μs) is longer than that under 450 nm excitation (1.58 μs) (Supplementary Fig. 5b and Supplementary Table 5), which is attributed to the more contribution of long-lived emitters to average decay time under 340 nm excitation.

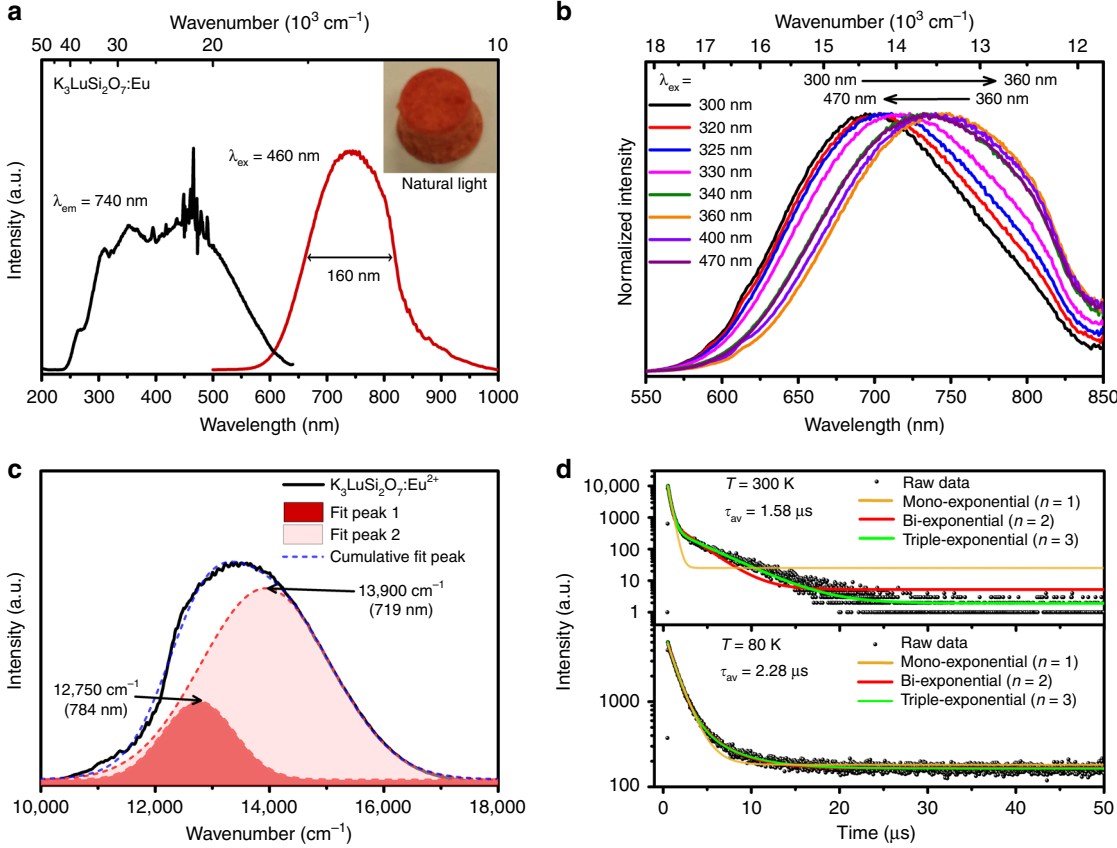

**Fig. 2** Photoluminescence properties of $K_3LuSi_2O_7:Eu^{2+}$ phosphors. **a** PL and PLE spectra of $K_3LuSi_2O_7:0.01Eu^{2+}$ measured/monitored at 460 and 740 nm, and the inset shows the photograph of $K_3LuSi_2O_7:0.01Eu^{2+}$ with deep red body color under natural light. **b** Room temperature PL spectra of $K_3LuSi_2O_7:0.01Eu^{2+}$ under different excitation wavelength from 300 to 470 nm. **c** The emission spectrum of the $K_3LuSi_2O_7:0.01Eu^{2+}$ and Gaussian fitting curves. **d** Decay curves and fitting results of $K_3LuSi_2O_7:0.01Eu$ measured at 80 K and 300 K under 450 nm pulse laser diode excitation

To evaluate the robustness of phosphors to heat and moisture, two sets of $K_3LuSi_2O_7:Eu$ samples were exposed to the extremely environmental condition at 80% relative humidity and 80 °C. The results suggest that the chemical stability of $K_3LuSi_2O_7:Eu^{2+}$ is relatively good (see details in Supplementary Fig. 6). The $Eu^{2+}$ content rarely changed to $Eu^{3+}$ depending on the prolonging duration time suggesting the good stability. One can also note that a small amount of $Eu^{3+}$ can be detected for the freshly prepared materials (Supplementary Fig. 7). Samples with different $Eu^{2+}$-doping concentrations ($x = 0.005$–$0.05$) have been prepared to optimize the PL intensity of $K_3LuSi_2O_7:xEu^{2+}$. As displayed in Supplementary Fig. 8a, the emission intensities increase slightly till up to $x = 0.01$ and then decrease with further increase in $x$. Based on the Dexter's theory and the Eu-Eu distances in $K_3LuSi_2O_7:Eu^{2+}$[34], energy transfer among the nearest neighbor ions accounts for the observed concentration quenching (see details in Supplementary Fig. 8a). The internal quantum efficiency of selected $K_3LuSi_2O_7:0.01Eu^{2+}$ is measured to be about 15% ($\lambda_{ex} = 460$ nm). Furthermore, one can find that the normalized emission peaks of $K_3LuSi_2O_7:xEu^{2+}$ show no shift as $Eu^{2+}$ doping content increase (Supplementary Fig. 8b).

**Mechanism of NIR emission in $K_3LuSi_2O_7:Eu^{2+}$.** First of all, we need to correlate the $Eu^{2+}$ site contribution and different emission center by considering local environments of activators. The $Eu^{2+}$ emission is dependent on the CFS. In general, when the average bond length ($d_{av}$) is shorter and the distortion indices ($D$) is larger, then there will be a larger CFS[35]. The average bond lengths, $d$(Lu-O) and $d$(K2-O) in $K_3LuSi_2O_7$ are 2.22 Å and

2.77 Å, respectively. The distortion indices are 0 for both $LuO_6$ and $K2O_6$ polyhedron. Thus, we believe that the higher energy emission (719 nm) peak is associated with $Eu^{2+}$ occupying K2 sites, while the lower energy emission (784 nm) peak is attributed to $Eu^{2+}$ occupying Lu sites. The conclusions of the site occupation for $Eu^{2+}$ in $K_3LuSi_2O_7$ are consistent with the results in $Rb_3YSi_2O_7:Eu^{2+}$ phosphor reported by our group[31].

It is well-known that the free $Eu^{2+}$ ions possess a large energy gap (4.2 eV) between the 5d excited state and the 4f ground state[36]. Once $Eu^{2+}$ are introduced into a given compound, the resulting emission wavelengths are affected by centroid shift, crystal splitting, and Stokes shift effects[37]. For $K_3LuSi_2O_7:Eu^{2+}$ phosphor, the centroid shift and Stokes shift have weak effect on the redshift because of the low covalence between $Eu^{2+}$ and $O^{2-}$ ions and the highly symmetric $LuO_6$ and $K2O_6$ polyhedrons, respectively[38,39]. The CFS depends especially on the coordination numbers, distortion index and size of the polyhedrons[28]. The substitution of $Eu^{2+}$ ions in $LuO_6$ and $K2O_6$ polyhedrons, and thus a design principle of selective occupation of $Eu^{2+}$ at small coordination numbers takes effect. It results in large CFS, which in turn leads to a large redshift. A comparison of the local environment of $Eu^{2+}$ ions in $K_3LuSi_2O_7$ and $Rb_3YSi_2O_7$ benefits further understanding of the abnormal NIR emission in $K_3LuSi_2O_7$. Firstly, they are isomorphic, and $Eu^{2+}$ ions occupy the same crystallographic lattices in both materials, and the distortion index of $YO_6/LuO_6$ and $K2O_6$ polyhedron are all 0[31]. Thus, the effect of polyhedral distortion on the NIR emission can be neglected. Secondly, the $d_{av}$ values of Lu-O and K2-O in $K_3LuSi_2O_7$ are 2.22 Å and 2.77 Å, respectively, which are much smaller than that of Y-O and K2-O (2.29 Å and 2.86 Å) in $Rb_3YSi_2O_7$, resulting in a larger CFS in 5d energy levels

and a larger redshift[40]. Therefore, the NIR light emission (740 nm) in $K_3LuSi_2O_7$:$Eu^{2+}$ is reasonable compared with the red emission (622 nm) in $Rb_3YSi_2O_7$:$Eu^{2+}$. In summary, the selective occupation of $Eu^{2+}$ ions in polyhedrons with small coordination numbers and small average bond length contribute to the NIR emission and the broad excitation band in $K_3LuSi_2O_7$:$Eu^{2+}$.

**Thermal quenching of PL.** Normally, the thermal effect of the NIR phosphor is crucial for the LED application, and thus the thermal quenching behavior of $K_3LuSi_2O_7$:$Eu^{2+}$ was investigated at different temperature from 80 to 500 K. Figure 3a demonstrates that the emission intensity decreases slowly with the increase of temperature owing to the enhanced nonradiative transition probabilities. As given in Fig. 3b, the integrated intensity at 150 °C retains ~59% of that at room-temperature. According to Dorenbos's model, the activation energy $\Delta E$ was calculated by using the following empirical formula:[40,41]

$$\Delta E = \frac{T_{0.5}}{680} eV \qquad (2)$$

where $T_{0.5}$ denotes the quenching temperature, and it is defined as the temperature when the emission declines to half of its initial intensity. Here, $\Delta E$ is calculated to be ~0.65 eV.

One can find from Fig. 3a that the emission peaks shift to shorter wavelength (blueshift) at evaluated temperature from 300 to 500 K. Furthermore, the temperature-dependent PL spectra under excitation of 460 nm of $K_3LuSi_2O_7$:$0.01Eu^{2+}$ were measured from 80 to 500 K, and the normalized spectra are presented in Fig. 3c. The emission peaks demonstrate a remarkable blueshift from 790 nm (80 K) to 710 nm (500 K) with increasing temperature, and the FWHM of emission bands gradually broadens with the increase of temperature. These observations are associated with the different

response of $Eu^{2+}$ ions located at the Lu and K2 sites to the temperature. In addition, the emission peaks at high temperature (790 nm) and low temperature (710 nm) are close to the two Gaussian fitted peaks at room temperature (784 nm, 719 nm) in Fig. 2c. This further validates the feasibility of the previous analysis about the site occupation of $Eu^{2+}$ ions.

Thus, a simple schematic $Eu^{2+}$ energy level diagram in $K_3LuSi_2O_7$ host is depicted in Fig. 3d. Based on the thermal ionization model proposed by Dorenbos, the smaller the gap (photoionization barrier $\Delta E_A$) between the bottom of conduction band and the upper edge of 5d energy level, the greater is the thermal quenching[40]. In $K_3LuSi_2O_7$:$Eu^{2+}$, the larger CFS of $Eu^{2+}$ occupied in Lu site produces a smaller $\Delta E_A$, and thus a poor thermal stability compared with $Eu^{2+}$ in the K2 site (Fig. 3d). Similarly, the larger CFS of $Eu^{2+}$ in $K_3LuSi_2O_7$ results in a smaller $\Delta E_A$, and thus it enables a poor thermal stability compared with that of $Rb_3YSi_2O_7$:$Eu^{2+}$[31]. Therefore, the blueshift occurred when heated. At low temperature of 80 K, the thermal ionization of 5d electrons becomes weak, and the dominant emission peak centers at longer wavelength (790 nm) associated with the more contribution from $Eu^{2+}$ ions occupying Lu sites. In contrast, thermal quenching behavior at high temperatures has a much weaker effect on the $Eu^{2+}$ ions in K2 sites than on the $Eu^{2+}$ ions in Lu sites, resulting in a relatively stronger emission at short wavelength. Therefore, with temperature increasing, the blueshift and the broadening of emission spectra occur simultaneously.

**NIR pc-LED device and applications.** A pc-LED with NIR emission was fabricated by combining $K_3LuSi_2O_7$:$Eu^{2+}$ phosphor and the commercial blue light-emitting InGaN chip (460 nm). The emission spectrum of the lighted lamp was monitored by a

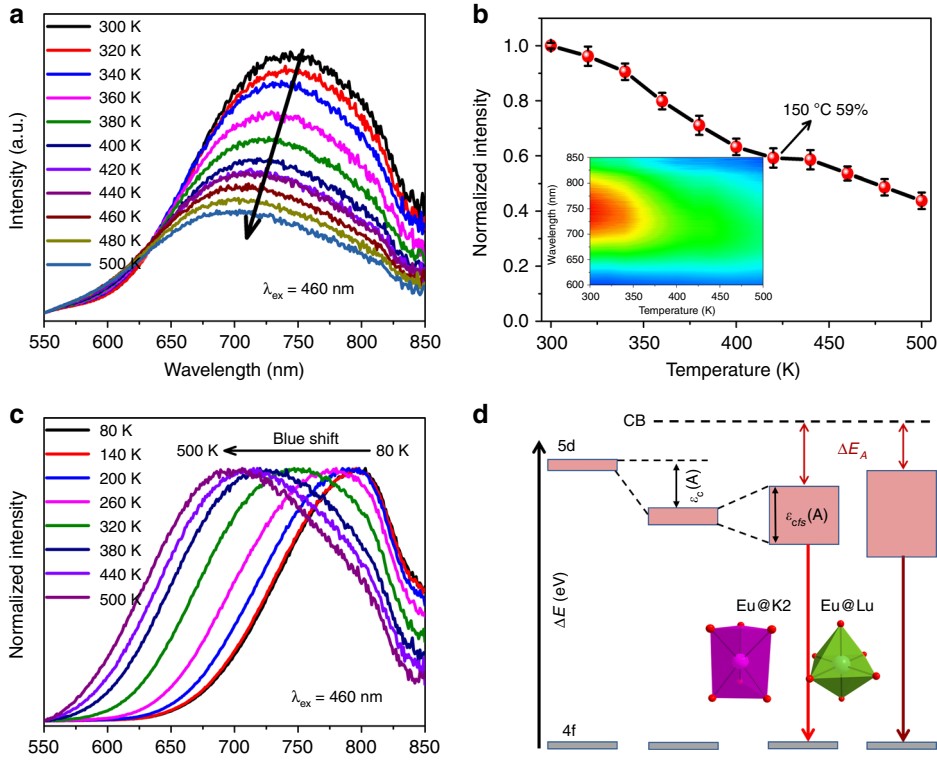

**Fig. 3** Thermal quenching properties of $K_3LuSi_2O_7$:$Eu^{2+}$ phosphor. **a** PL spectra and **b** normalized emission intensities of $K_3LuSi_2O_7$:$0.01Eu^{2+}$ as a function of temperature in 300–500 K under 460 nm excitation. **c** Normalized temperature-dependent PL spectra in 80–500 K at intervals of 60 K. **d** Schematic energy level diagram of $Eu^{2+}$ in $K_3LuSi_2O_7$. $\varepsilon_c(A)$ denotes the centroid shift of 5d-levels of $Eu^{2+}$, $\varepsilon_{cfs}(A)$ shows the CFS effect in connection with coordination number of cations and polyhedrons' size, $\Delta E_A$ represents the photoionization barrier

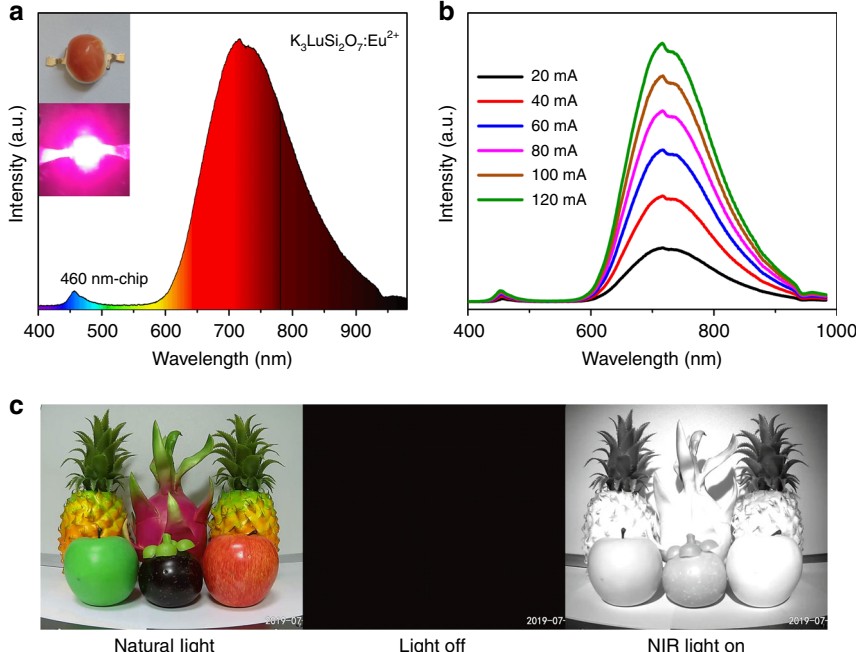

**Fig. 4** NIR application of $K_3LuSi_2O_7$:Eu based pc-LEDs. **a** Photographs and PL spectrum of as-fabricated pc-LED combining a commercial blue LED and $K_3LuSi_2O_7$:0.01Eu$^{2+}$ phosphor. **b** PL spectra of the pc-LED upon various forward bias currents. **c** Photographs under natural light and NIR pc-LED light captured by the corresponding visible camera and a NIR camera

visible-NIR continuous fiber spectrophotometer. The broad-band PL spectrum and the as-obtained and lighted NIR pc-LED lamps are given in Fig. 4a. The slight spectral deviation from the lamp and the powder is related with the calibration difference of the used spectrometers. Figure 4b demonstrates the PL spectra of LED device upon different forward bias currents (20–120 mA). The NIR emission output power increase with the increase of forward currents and reaches 21.5 mW when the forward current is 100 mA (Supplementary Table 6). Figure 4c shows the photographs obtained by different cameras under natural light and NIR pc-LED light, respectively. Nothing can be detected by NIR camera once the NIR pc-LED is off. In contrast, the NIR camera can capture black-and-white images while the NIR pc-LED lamp is lighted. These results indicate the application of the $K_3LuSi_2O_7$:Eu$^{2+}$ phosphor in night-vision technologies, and other potential fields including medical and food industries can be also expected for such NIR phosphors.

## Discussion

To conclude, we have designed and successfully synthesized the Eu$^{2+}$-activated broad-band-emitting NIR phosphor with super-broad excitation band from 250 to 600 nm. Under 460 nm blue light excitation, $K_3LuSi_2O_7$:Eu$^{2+}$ shows an unprecedented broad NIR emission band peaking at 740 nm and also demonstrates a high FHWM of 160 nm. Eu$^{2+}$ selectively occupancy at Lu and K2 sites, and the small coordination numbers and small average bond lengths lead to the as-observed NIR emission with characteristic $4f$-$5d$ transition of Eu$^{2+}$. The as-fabricated NIR pc-LEDs based on $K_3LuSi_2O_7$:Eu$^{2+}$ phosphors have been applied in night-vision devices and demonstrated potential applications. This finding gives the feasibility of the feasible strategy to discover Eu$^{2+}$ doped NIR phosphor with large redshift by occupying polyhedrons with small coordination numbers. More importantly, it initiates a new way to explore NIR phosphors, i.e. achieve NIR emission by Eu$^{2+}$ doping in the solid-state materials.

## Methods

**Materials and synthesis of the NIR phosphor**. $K_3LuSi_2O_7$:$x$Eu$^{2+}$ phosphors with different Eu$^{2+}$ content ($x = 0$–$0.05$) were prepared by employing solid-state reactions. The stoichiometric mixtures of $K_2CO_3$ (A.R.), $Lu_2O_3$ (99.99%), $SiO_2$ (A.R.), and $Eu_2O_3$ (99.99%) were thoroughly grounded and transferred into an alumina crucible. After that, the mixed starting materials were sintered at 1350 °C for 6 h in a forming gas atmosphere ($N_2$:$H_2 = 80$:20), and finally quenched to room temperature. As a typical synthesis, 3.8 g raw materials ($K_2CO_3$, $Lu_2O_3$, $SiO_2$, and $Eu_2O_3$) in the given stoichiometric ratio can produce ~2.3 g products.

**Characterization**. A D8 Advance X-ray diffractometer (XRD, Bruker Corporation, Germany) was used to collect the XRD patterns, and the operation voltage and current were set as 40 kV and 40 mA, respectively. Structural characterization and Rietveld refinement were conducted using TOPAS 4.2 software[42]. Diffuse reflectance spectra were collected using a Hitachi UH4150 ultraviolet-visible-near infrared spectrophotometer. Photoluminescence emission and excitation spectra were measured by a FLSP920 fluorescence spectrophotometer, and both visible and NIR PMT detectors were applied. PL spectra of $K_3LuSi_2O_7$:Eu$^{2+}$ and as-fabricated NIR pc-LED were comparatively measured by a fiber spectrophotometer (NOVA high sensitive spectrometer, idea optics, China), and the 375-nm and 450-nm laser diode acted as the excitation source. The temperature-dependent spectra were evaluated by a FLS920 spectrophotometer equipped with a MercuryiTc temperature control instrument (OXFORD). Liquid nitrogen was used to cool temperature. The decay time was measured by a FLS920 instrument equipped with 450 nm and 340 nm pulse laser diode. The statistical photons are 5000 and 10,000, respectively, for the measurements at low temperature (80 K) and room temperature (300 K), and the decay curvers were fitted by the FAST software attached with FLS920 spectrophotometer. All the photophysics properties measurements were repeated at least three times to confirm the data. The quantum efficiency was measured using a Hamamatsu absolute photoluminescence quantum yield spectrometer C11347 Quantaurus-QY. The photoelectric properties and emission spectra of the fabricated pc-LEDs were measured by a Morpho 3.2 software equipped with a NOVA Laboratory Class Spectrometer. Photographs for the application of NIR pc-LEDs are taken by a NIR and a visible camera.

## Data availability

The data that support the plots within this paper and other findings of this study are available from the corresponding author upon reasonable request.

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

## Acknowledgements

This work is supported by the National Natural Science Foundation of China (Nos.: 51722202 and 51972118), Fundamental Research Funds for the Central Universities (D2190980), and the Guangdong Provincial Science & Technology Project (2018A050506004).

## Author contributions

Z.X. initiated and guided the research. J.Q. synthesized the samples and wrote the manuscript, and Z.X. and Q.Z. revised it. G.Z. and Y.Z. discussed the luminescence properties and made the NIR pc-LED device. All authors discussed the results and commented on the manuscript.

## Competing interests

The authors declare no competing interests.
