## [Peer Review File · Nature Communications]

Reviewers' comments:

Reviewer #1 (Remarks to the Author):

This is an interesting paper reporting novel near infrared (NIR) luminescent phosphor solid solutions, $K_3LuSi_2O_7:Eu^{2+}$ exhibiting super broad excitation bands. Pure phase of the reported materials were successfully synthesized by solid-state reactions and the structures were determined by powder X-ray diffraction using Rietveld refinements. Photoluminescent properties and the structure-property relationships were nicely reported by the authors. Therefore, the work can be acceptable to Nat. Commun. A couple of suggestions:

Commun. A couple of suggestions:

- Current structural models refined by the Rietveld refinement need to be modified. In the reported model, Eu occupies only the site of Lu. However, the authors explain that Eu occupies to the sites of Lu and K. The refinements need to be re-done for the reported compounds.
- The section name 'Discussion' should be 'Conclusions'.

Reviewer #2 (Remarks to the Author):

1) Key results: Please summarise what you consider to be the outstanding features of the work.

The authors hypothesized that the known dependence of Eu(II) spectroscopy on the ion's coordination geometry could be exploited to shift the materials' emission spectrum towards the red by replacing the larger Rb(I) with smaller K(I) in the previously reported $Rb_3YSi_2O_7$, thus providing access to much-coveted blue-excitable, broadly emissive NIR-LEDs. Their hypothesis was validated, and materials with broad emissions above 700 nm were obtained. The rationale for the red-shift was investigated, and found to be consistent with shorter bond distances in the NIR-emitting K-based materials compared to the more blue-emitting Rb-based ones.

One of the most attractive aspects of the work is the simplicity and elegance of the hypothesis, and the straightforward way the new materials could be derived from existing ones.

On a more fundamental note, it is impressive that the Eu(II) emission can be tuned to such an extent by a simple replacement of a reagent. This is an effect that is usually much better studied in discrete molecules, where however the synthetic costs tend to be huge. The authors should probably emphasize this aspect of the work a bit more.

2) Validity: Does the manuscript have flaws which should prohibit its publication? If so, please provide details.

There are no such flaws in the sections which I was competent to evaluate. However, please see comments below on the photophysics part.

3) Originality and significance: If the conclusions are not original, please provide relevant references. On a more subjective note, do you feel that the results presented are of immediate interest to many people in your own discipline, and/or to people from several disciplines?

The work is a logical continuation of previous results in the field, including work by the authors themselves, and it has delivered excellent results. (By 'logical' I do not mean unoriginal, but that it was a step that had to be taken by someone.)

A blue-emitting NIR LED is very exciting, and I would expect many scientists in my area to be interested in these results.

4) Data & methodology: Please comment on the validity of the approach, quality of the data and

quality of presentation. Is the reporting of data and methodology sufficiently detailed and transparent to enable reproducing the results?

The study is well planned and the data are clearly presented. The data shown support the authors' conclusions.

There are a number of additional experiments that could strengthen the work, please see the these below. Furthermore:

- There are a number of details in the Experimental for the synthesis and the photophysical studies that should be added for the experiments to be reliably reproducible (see below).
- Please provide the residuals of the fit for the lifetime data (possibly in the supporting information). How do alternative fits (mono- and multiexponential ones) compare to the biexponential fit?

5) Appropriate use of statistics and treatment of uncertainties: All error bars should be defined in the corresponding figure legends; please comment if that's not the case. Please include in your report a specific comment on the appropriateness of any statistical tests, and the accuracy of the description of any error bars and probability values.

- Error bars are not provided in Figure 3b. It is unclear how many times the emission lifetimes were measured. This, and the standard deviation on the lifetimes should be provided.
- What is the uncertainty (error) on the amount of Eu dopant in the materials? What is the real difference between $x = 0.005$ and $x = 0.01$? It would be helpful to provide the comparison of the photophysical properties of two different batches of sample with e.g. $x = 0.005$ to demonstrate the reproducibility of the process.
- The bond lengths are given as e.g. 2.2897 \AA , which is too precise.

6) Conclusions: Do you find that the conclusions and data interpretation are robust, valid and reliable?

Largely, yes. Please see additional suggested experiments below. Also, the discussion of the thermal stability is somewhat misleading, as there are materials that are more stable. This section should be rephrased.

The thermal quenching data should be put in a bit more context. The materials are clearly more sensitive to an increased temperature than previously reported NIR-emitting LEDs (ref 28), and even the Rb-based analogues (80% retained at $300 \text{ }^\circ\text{C}$). As the authors provide a convincing explanation for the large thermal quenching, I do not think that this detracts from the work, but being explicit about these would help non-specialists.

7) Suggested improvements: Please list additional experiments or data that could help strengthening the work in a revision.

- Please comment on the chemical stabilities of the materials. How robust are they to air or moisture? Are there any Eu(III) ions in the lattice in the freshly prepared materials? Is there any sign of Eu(II) over time?
- Above the $x = 0.01$ Eu level the emissiveness of the materials decrease. The authors suggest that this is due to concentration quenching. This statement should be supported by a calculation of the average Eu-Eu distances at the various doping levels. The dilution of the Eu(II) dopant with an appropriate spectroscopically silent analogue may be possible, although that would require a full characterization of the new materials.
- And as a follow-up to the previous questions: can the quenching at high Eu levels be due to the presence of Eu(III) contaminants? These may be difficult to see in steady-state mode, but their presence may be probed by time-resolved experiments with longer delays and direct excitation at 394 nm of a putative Eu(III) center.

- The authors suggest that there are two non-identical emissive Eu(II) centers in the materials. This conclusion is supported by several data. However, it would be interesting to see the lifetime decay data for some other doping levels – preferably at least two more. Do the relative contributions of the short- and long-lived emitters change with doping levels?
- How do the short- and long-lived emitter contributions change upon changing the excitation wavelength?

8) References: Does this manuscript reference previous literature appropriately? If not, what references should be included or excluded?

The references are appropriate. The ones cited in the Introduction provide a fair overview of the field and are representative of the state of the art. In general, the authors provide good supporting references for their statements. The only exception is the discussion on the effects of coordination geometry on the Eu(II) spectroscopy (in the 'Mechanism of NIR emission...'), where the statement on the average bond length and distortion should be supported by a reference. (bottom of page 3) On a different note, a number of the cited references are missing page numbers and/or DOIs. This is presumably an EndNote problem, but should be fixed.

9) Clarity and context: Is the abstract clear, accessible? Are abstract, introduction and conclusions appropriate?

The abstract should be somewhat re-written, as the first half reads more like an abbreviated introduction. Specifically, references to Ce³⁺ should be removed, as it is not used in the work. Similarly the text on the strong nephelauxetic effect and crystal field splitting, as it is too technical. The term 'abnormal NIR emission' should be replaced with 'unusual NIR emission', or 'unexpected NIR emission', or even better, 'unprecedented NIR emission'; as the emission is well explained in the manuscript, so is not abnormal (in fact, in line with the authors' hypothesis).

The introduction and the conclusions are appropriate in terms of general content. I have the following minor comments about the text:

- The first sentence is too long, and is too general. It should be broken up into 2-3 shorter sentences. Furthermore, the end 'and the unique NIR absorption, etc.' is too vague.
- There are a number of instances where subject and verb are unmatched, i.e. subjects in singular are paired with verbs in plural and vice versa.
- A few words that are too colloquial. E.g.: 'famous activators' (page 1), 'it is easy to understand' (page 2), 'super broad excitation' (several instances).
- Figure 2 is not referenced in the text. The references made to Figure 1 in the Photoluminescence properties section should be to Figure 2.
- On page 3, in column, the occupation mechanism is given. It is unclear what the different symbols mean in the equation. What are Eu^{*}K, Eu^{*}Y, etc? Are these really Y?
- Please round the wavenumbers to the nearest 50 cm⁻¹.
- Please round the lifetimes as appropriate for the experimental setup.
- In figures with spectra, please provide a scale with both nm and cm⁻¹ (possibly with nm at the bottom and cm⁻¹ at the top).
- In the legend of Figure S3 it should be 'reflectance' and not 'reflection'.
- In Figure 3c 'Blue shif' should be 'Blue shift'
- Please provide details on how the intensity was normalized in Figure S3b. Given the noise in the spectra, it would be helpful to provide the ratio of the 300 and 440 nm (or some wavelengths where the data are not as noisy) at the various emission wavelengths.
- On page 2, in the discussion of the crystal structure: the sizes of the ions are compared, without giving the actual values. It would be important to have the ionic radii of K⁺, Eu²⁺, Lu³⁺ and Rb⁺ in the text for ready comparison.

10) Please indicate any particular part of the manuscript, data, or analyses that you feel is outside the scope of your expertise, or that you were unable to assess fully.

I am not a structural chemist, and am not competent to evaluate the quality of the XRD data.

11) Is the manuscript clearly written? If not, how could it be made more clear or accessible to nonspecialists?

Yes. The abstract and the introduction should be slightly rephrased e.g. as suggested above.

12) Would readers outside the discipline benefit from a schematic of the main result to accompany publication?

Possibly.

13) Could the manuscript be shortened? (Because of pressure on space in our printed pages we aim to publish manuscripts as short as is consistent with a persuasive message.)

The manuscript is quite concise, and I'm reluctant to suggest substantial shortening. However, the fact that there are two distinct emitters in the materials is discussed in several places and in different contexts. It should be possible to combine these into a shorter section.

14) Should the authors be asked to provide supplementary methods or data to accompany the paper online?

They should provide more data on the synthesis and the handling of the isolated materials. For example the scale of the syntheses, the number of times they were repeated, the yield of product, isolation. This is currently missing, and it makes the synthetic protocols irreproducible.

15) Have the authors done themselves justice without overselling their claims?

Yes.

16) Have they been fair in their treatment of previous literature?

Yes.

17) Have they provided sufficient methodological detail that the experiments could be reproduced?

Please see the comment above on the synthesis. I'd also prefer to see more detail on the photophysical studies, although there is quite a bit of information in the figure legends. Specifically, the variable temperature experiments should be discussed (how were these performed, how was cooling provided, etc). How was the lifetime fitting performed? Which software was used for data acquisition and processing?

18) Is the statistical analysis of the data sound, and does it conform to the journal's guidelines?

Not generally applicable, but see comment on the standard deviation of the lifetime data.

19) Are the reagents generally available?

Yes.

20) Are there any special ethical concerns arising from the use of human or other animal subjects?

No.

Reviewer #3 (Remarks to the Author):

Manuscript under the title "Eu²⁺-Doped Near-Infrared-Emitting Phosphor for Light-Emitting Diodes" reports synthesis and optical properties of the target materials. From the application point of view, the material is very interesting. However, there are some major issues in the manuscript, that authors should take care of.

1. The English should be carefully reinspected, since there are many grammar and punctuation mistakes. Besides, some figures in the manuscript are assigned incorrectly.

2. Authors state that Eu²⁺ ions occupies two different sites, namely, K₂O₆ and LuO₆. However, Rietveld refinement data presented in Table S2 show, that the refinement was performed only as Eu²⁺ ions were occupying Lu³⁺ sites. This contradicts the discussion in the manuscript.

3. My major concern is with the spectroscopic data of the synthesized materials:

a) Fig. 2a shows the excitation spectrum of K₃LuSi₂O₇:0.01%Eu²⁺ compound. However, there are some sharp lines ca. 450 nm. Where these lines come from. Eu²⁺ ions possess no sharp lines in excitation spectra, thus it should be instrument/measurement related. Were excitation spectra corrected for instrument response?

b) The reflection spectra given in Fig. S3a are also rather unusual, especially the one for undoped sample. It is strange, that the reflectance of undoped sample is the highest at ca. 250 nm and then decreases. More commonly, the results are opposite, i.e., the reflectance at short wavelengths for undoped materials is lower if compared to longer wavelengths. Besides, what is the bandgap of undoped material? The reflectance values of the synthesized samples at longer wavelengths is ca. 90%. This usually suggests that some defects are found in the material. What was the body colour of the synthesized materials? Was it greyish?

c) Fig. 2c shows the emission spectra fit with two Gaussian curves. Since the wavelength was converted to energy scale, were the intensity also corrected for such conversion? (see, for instance, 10.1016/j.optmat.2012.01.034).

d) The PL lifetime values of Eu²⁺ doped phosphors in the red spectral region usually are around 1 μs. However, the authors report significantly lower values. What could be the reason for such mismatch?

e) Authors mentioned that two detectors were employed in PL spectra measurements, i.e., VIS and NIR. The excitation wavelength and temperature dependent emission spectra were obviously recorded using VIS detector. The sensitivity of VIS detector in the red/infrared part of the spectrum is rather low and thus the correction for instrument response (I assume authors have done that since Edinburgh Instruments spectrometers always come with emission correction files) will significantly increase the intensity in this area especially if the intensity of the sample is relatively low in this area as well. Besides, the spectra measured with this detector is not complete. Taking into account that temperature dependent emission integrals were also analyzed in Fig. 3b the significant part of emission data are not present in this figure. Therefore, the temperature dependent and excitation wavelength dependent emission should also be measured with NIR detector.

f) Authors also discuss the activation energy of thermal quenching induced by photoionization referring to model proposed by P. Dorenbos. However, even though the temperature dependent integrated emission data are given, the activation energy was not calculated. Why?

Answers to Reviewer #1 Comments:

This is an interesting paper reporting novel near infrared (NIR) luminescent phosphor solid solutions, $K_3LuSi_2O_7:Eu^{2+}$ exhibiting super broad excitation bands. Pure phase of the reported materials were successfully synthesized by solid-state reactions and the structures were determined by powder X-ray diffraction using Rietveld refinements. Photoluminescent properties and the structure-property relationships were nicely reported by the authors. Therefore, the work can be acceptable to Nat. Commun.. A couple of suggestions:

Current structural models refined by the Rietveld refinement need to be modified. In the reported model, Eu occupies only the site of Lu. However, the authors explain that Eu occupies to the sites of Lu and K. The refinements need to be re-done for the reported compounds. The section name ‘Discussion’ should be ‘Conclusions’.

Author reply: We are very grateful to the reviewer for his/her comment. As for the description about the site occupation of Eu^{2+} in the structural Rietveld refinement, possibly we don’t express our result in a proper way and mislead the reviewer. In fact, we can infer that Eu^{2+} occupy both K and Lu sites from Rietveld refinement results. Firstly, if Eu^{2+} only occupy small Lu sites, the cell volume will show a linear increasing trend with the increase of Eu doping concentration. However, the nonlinear increasing in Reference **Fig. 1a** indicates that Eu also occupy the large K ions. Secondly, the bond length of K2-O and Lu-O changed with the doping concentration Reference **Fig. 1b**, which further certify that Eu occupy K2 and Lu sites in $K_3LuSi_2O_7$. Anyway, to avoid misleading readers, we have performed the Rietveld refinement again and deleted the sites occupancy of Eu ions in Lu sites, and the result is shown in **Table S2** in the revised SI.

In addition, we have revised the section name “Discussion” to “Conclusions” in the revised manuscript. Thanks.

Reference Fig. 1 a Cell volume $V(x)$ dependence of the Eu doping concentration ($x = 0, 0.01, 0.02, 0.04$). **b** Dependence of the average bond length of Lu, K1, and K2 in $K_3LuSi_2O_7:xEu^{2+}$ ($x = 0, 0.01, 0.02, 0.04$).

Answers to Reviewer #2 Comments:

Question 1

Key results: Please summarize what you consider to be the outstanding features of the work.

The authors hypothesized that the known dependence of Eu(II) spectroscopy on the ion's coordination geometry could be exploited to shift the materials' emission spectrum towards the red by replacing the larger Rb(I) with smaller K(I) in the previously reported $\text{Rb}_3\text{YSi}_2\text{O}_7$, thus providing access to much-coveted blue-excitable, broadly emissive NIR-LEDs. Their hypothesis was validated, and materials with broad emissions above 700 nm were obtained. The rationale for the red-shift was investigated, and found to be consistent with shorter bond distances in the NIR-emitting K-based materials compared to the more blue-emitting Rb-based ones.

One of the most attractive aspects of the work is the simplicity and elegance of the hypothesis, and the straightforward way the new materials could be derived from existing ones.

On a more fundamental note, it is impressive that the Eu(II) emission can be tuned to such an extent by a simple replacement of a reagent. This is an effect that is usually much better studied in discrete molecules, where however the synthetic costs tend to be huge. The authors should probably emphasize this aspect of the work a bit more.

Author reply: We are grateful to the reviewer for the positive evaluation of our manuscript.

Yes, the synthesis cost of inorganic phosphor is low compared with the other luminescence materials and the strategy to fabricate the LED. We have added related description in the introduction part (see page 1).

Question 2

Validity: Does the manuscript have flaws which should prohibit its publication? If so, please provide details.

There are no such flaws in the sections which I was competent to evaluate. However, please see comments below on the photophysics part.

Author reply: Thanks for the positive comments for the publication. We have also responded the comments on the photophysics part.

Question 3

Originality and significance: If the conclusions are not original, please provide relevant references.

On a more subjective note, do you feel that the results presented are of immediate interest to many people in your own discipline, and/or to people from several disciplines?

The work is a logical continuation of previous results in the field, including work by the authors themselves, and it has delivered excellent results. (By 'logical' I do not mean unoriginal, but that it was a step that had to be taken by someone.)

A blue-emitting NIR LED is very exciting, and I would expect many scientists in my area to be interested in these results.

Author reply: Thanks for the positive comments.

Question 4

Data & methodology: Please comment on the validity of the approach, quality of the data and quality of presentation. Is the reporting of data and methodology sufficiently detailed and transparent to enable reproducing the results?

The study is well planned and the data are clearly presented. The data shown support the authors' conclusions.

There are a number of additional experiments that could strengthen the work, please see these below. Furthermore: There are a number of details in the Experimental for the synthesis and the photophysical studies that should be added for the experiments to be reliably reproducible (see below). Please provide the residuals of the fit for the lifetime data (possibly in the supporting information). How do alternative fits (mono- and multiexponential ones) compare to the biexponential fit?

Author reply: Thanks for your suggestions.

Firstly, we have added additional experiments details in the experimental section according to the suggestions of the reviewer.

Secondly, it is indeed very important on the details of the photophysical studies. For example, the cutoff command during decay curve measurement is previously set to the test time (4 minutes). However, this time is too short to get enough photons to obtain accurate lifetime values, especially for the samples with weak emission intensity. Therefore, such a method will mislead the readers indeed. During the revision, we measured lifetime values by using the same FLS920 instrument equipped with 340 and 450 nm pulse laser diodes as the excitation source. The new method is based on the statistical photons of 5000 and 10000, respectively, for the measurements at low temperature (80 K) and room temperature (300 K), respectively. All the data were fitted by the FAST software attached with FLS920. The analysis details are given below and the corresponding contents have been also updated in the revised manuscript.

Reference **Fig. 2** shows the decay curves measured at 80 K and 300 K under 450 nm pulse laser diode excitation. Obviously, the decay curves measured at 80 K can be well fitted by both the bi- and triple-exponential, and two fitting curves nearly overlap. This result further demonstrate there are two luminescent centers in $K_3LuSi_2O_7$. However, the decay curves at 300 K can be only well fitted by the triple-exponential, which is ascribed to an additional nonradiative transition at room temperature originating from the lattice vibration. Therefore, the decay time value measured at 300 K (1.58 μ s) is shorter than that at 80 K (2.28 μ s). The detailed fitting parameters are listed in reference **Table 1** below.

Reference Fig. 2 The decay curves and fitting profiles of $K_3LuSi_2O_7:0.01Eu$ measured at 80 K and 300 K under 450 nm pulse laser diode excitation.

According to the suggestion of the reviewer, some experimental and calculation details are listed below and also given in the SI file during this revision,

Triple-exponential fitting functions:¹

$$I = A_1 \exp\left(-\frac{t}{\tau_1}\right) + A_2 \exp\left(-\frac{t}{\tau_2}\right) + A_3 \exp\left(-\frac{t}{\tau_3}\right) \quad (1)$$

Average decay times are calculated by the following question:

$$\tau_{av} = \frac{A_1\tau_1^2 + A_2\tau_2^2 + A_3\tau_3^2}{A_1\tau_1 + A_2\tau_2 + A_3\tau_3} \quad (2)$$

where τ_1 , τ_2 and τ_3 are lifetimes for different components; A_1 , A_2 and A_3 are the corresponding fitting constants.

Reference Table 1. Lists of the luminescent decay times (τ_1 , τ_2 , τ_3), fitting constants (A_1 , A_2 , A_3) average decay times (τ_{av}) and standard error of τ_{av} during the fitting of the decay curves of $K_3LuSi_2O_7:Eu$ samples with different Eu^{2+} contents, which are measured at 80 K and 300 K under 450 nm and 340 nm pulse laser diodes excitation. All the data were fitted by the FAST software attached with FLS920.

$\lambda_{ex}=450\text{ nm}, \lambda_{em}=740\text{ nm}, \text{Temperature}=300\text{ K}$								
x	A_1	τ_1	A_2	τ_2	A_3	τ_3	τ_{av} (μs)	τ_{av} Error
0.01	7538.88	0.22	1557.41	0.62	328.31	4.01	1.58	0.038
0.02	7870.61	0.24	1569.44	0.73	382.01	3.74	1.51	0.030
0.03	7200.87	0.21	1664.63	0.69	353.89	3.61	1.45	0.033
$\lambda_{ex}=450\text{ nm}, \lambda_{em}=740\text{ nm}, \text{Temperature}=80\text{ K}$								
0.01	1907.67	0.80	2519.20	1.55	312.02	5.35	2.28	0.025
$\lambda_{ex}=340\text{ nm}, \lambda_{em}=740\text{ nm}, \text{Temperature}=300\text{ K}$								
0.01	8120.81	0.22	1381.83	0.63	324.79	4.26	1.69	0.043

- The authors suggest that there are two non-identical emissive Eu(II) centers in the materials. This conclusion is supported by several data. However, it would be interesting to see the lifetime decay data for some other doping levels – preferably at least two more. Do the relative contributions of the short- and long-lived emitters change with doping levels?

- How do the short- and long-lived emitter contributions change upon changing the excitation wavelength?

Author reply: Thanks for the informative suggestions.

According to the suggestion of the reviewer, we added the additional measurement. The decay times will be shortened with increasing Eu content (reference **Fig. 3a** below) due to the energy transfer between Eu^{2+} luminescent centers. It is also found that the decay times of the long-lived emitters (τ_3) decreased with the doping concentration. Under 340 nm pulse laser diode excitation, the decay value is calculated to be 1.69 μs (reference **Fig. 3b**), which is longer than 1.58 μs under 450 nm excitation. This further certifies that there are different luminescent centers (short- and long-lived emitter) in the $K_3LuSi_2O_7:Eu$. The detailed fitting parameters are listed in reference **Table 1**. The decay times have been repeated at least three times to ensure the validity of the measurements.

Finally, to strengthen the contents and the results in this manuscript, we have added the related contents mentioned above in revised manuscript and supporting information during this revision.

Reference Fig. 3 **a** Decay curves and fitting results of $K_3LuSi_2O_7:xEu$ ($x = 0.01, 0.02, 0.03$) measured at 300 K under 450 nm pulse laser diode excitation. **b** Decay curves and fitting results of $K_3LuSi_2O_7:0.01Eu$ measured at 300 K under 450 nm and 340 nm pulse laser diode excitation.

Question 5

Appropriate use of statistics and treatment of uncertainties: All error bars should be defined in the corresponding figure legends; please comment if that's not the case. Please include in your report a specific comment on the appropriateness of any statistical tests, and the accuracy of the description of any error bars and probability values.

- Error bars are not provided in Figure 3b. It is unclear how many times the emission lifetimes were measured. This, and the standard deviation on the lifetimes should be provided.
- What is the uncertainty (error) on the amount of Eu dopant in the materials? What is the real difference between $x = 0.005$ and $x = 0.01$? It would be helpful to provide the comparison of the photophysical properties of two different batches of sample with e.g. $x = 0.005$ to demonstrate the reproducibility of the process.
- The bond lengths are given as e.g. 2.2897 \AA , which is too precise.

Author reply: Thanks for your informative suggestions.

According to the suggestion of the reviewer, the error bars have been added in **Figure 3b** (revised manuscript), and the standard errors of lifetimes are also provided in **Table S5** in the **SI** file. The bond lengths are round up after the third decimal point. As discussed previously, the decay time have been repeated at least three times to ensure the validity of the measurements. The standard deviation on the lifetimes have been added in the revised manuscript.

The Eu content is given according to stoichiometric ratios in the starting materials. Since the content is too low, one can hardly know the precise amount of Eu dopant in the final phosphor samples. There is also no precise experimental methods to determine the contents, moreover, this is normal not very important during the study since we can know the variation of the Eu contents for the photophysical properties.

Moreover, it is sure that the experiment results on photophysical properties are repeatable depending on different Eu contents. Reference **Fig. 4** shows the PL spectra of two batches of samples, which are designed with different Eu content and synthesized under the same condition. One can see that the emission intensity shows similarly change trend, and the optimal dopant concentration is 0.01.

The value for the bond length has been revised to be 2.29 \AA . And other values have been also

revised.

Reference Fig. 4 PL spectra of $\text{K}_3\text{LuSi}_2\text{O}_7:x\text{Eu}$ under 460 nm excitation, and (a) and (b) are listed as a comparison for the two batches of samples with different Eu content synthesized under the same condition.

Question 6

Conclusions: Do you find that the conclusions and data interpretation are robust, valid and reliable? Largely, yes. Please see additional suggested experiments below. Also, the discussion of the thermal stability is somewhat misleading, as there are materials that are more stable. This section should be rephrased. The thermal quenching data should be put in a bit more context. The materials are clearly more sensitive to an increased temperature than previously reported NIR-emitting LEDs (ref 28), and even the Rb-based analogues (80% retained at 300 °C). As the authors provide a convincing explanation for the large thermal quenching, I do not think that this detracts from the work, but being explicit about these would help non-specialists.

Author reply: Thanks for your informative suggestions.

The discussion of the thermal stability should give more information and description on the importance. Firstly, we reorganized some sentences in this section. Secondly, as also pointed out by the reviewer, the luminescence of $\text{K}_3\text{LuSi}_2\text{O}_7:\text{Eu}$ is relatively sensitive depending on an increasing temperature compared to that of the $\text{Rb}_3\text{YSi}_2\text{O}_7:\text{Eu}$. The reason is that, in $\text{K}_3\text{LuSi}_2\text{O}_7:\text{Eu}^{2+}$, the larger CFS of Eu^{2+} leads to a smaller ΔE_A , as given in the following reference **Fig. 5** below, and thus a poor thermal stability compared with the $\text{Rb}_3\text{YSi}_2\text{O}_7:\text{Eu}^{2+}$.

We have rephrased the corresponding description in the revised manuscript.

Reference Fig. 5 Schematic energy level diagram for Eu^{2+} ions in $\text{K}_3\text{LuSi}_2\text{O}_7:\text{Eu}$ and

Rb₃YSi₂O₇:Eu.

Question 7

Suggested improvements: Please list additional experiments or data that could help strengthening the work in a revision.

- Please comment on the chemical stabilities of the materials. How robust are they to air or moisture? Are there any Eu(III) ions in the lattice in the freshly prepared materials? Is there any sign of Eu(II) over time?

Author reply: Thanks for the comment.

We have added the additional experiment according to the suggestion. K₃LuSi₂O₇:Eu was exposed to the 80% relative humidity (RH) and 80 °C for different time to evaluate the robustness of samples to air and moisture. We carried out two sets of experiment, as shown in reference **Fig. 6a** and **Fig. 6b**, and the emission intensities decrease over times, and remain about 77% of the pristine sample after 240 min in such a harsh condition. The photographs (**Fig. 6c**) under natural light showed almost no changes for different treated samples. We can draw a conclusion that the chemical stabilities of the materials are relatively good and we have also added some descriptions on this in the revised manuscript.

Generally speaking, nearly all the Eu²⁺-containing samples possess Eu³⁺, which can be detected by the experiments (*Chem. Mater.* 2018, 30, 494-505.). For our samples, the answer is also “Yes”. There are some Eu³⁺ ions in the freshly prepared materials. Under 394 nm excitation with the characteristic transition of Eu³⁺, the intrinsic emission peaks (592 and 614 nm) of Eu³⁺ can be found for the freshly prepared K₃LuSi₂O₇:Eu (reference **Fig. 7a**). With the increase in exposure time in the air, only very small amount of Eu²⁺ will be oxidized to Eu³⁺. Therefore, the samples that exposed to the air for 60 days shows a relatively stronger emission of Eu³⁺ compared with the freshly prepared materials. Moreover, to further certify the existence of Eu³⁺, the time-resolved luminescence spectra (TRS) are measured in the time interval of 0-4 μs. Obviously, the emission peaks of Eu³⁺ co-existed in the emission peaks of Eu²⁺ when the transition time >2 μs (**Fig. 7b**).

In summary, the chemical stability of K₃LuSi₂O₇:Eu is relatively good and can be further improved by the surface coating as realized in other silicate samples. Eu³⁺ ions normally exist with Eu²⁺ for the freshly prepared K₃LuSi₂O₇:Eu samples, however, very small amount of them can be further transferred into Eu³⁺ depending on the prolonging duration time suggesting the stability.

We have added the corresponding description in revised manuscript and supporting information.

Reference Fig. 6 a (Experiment 1) and b (Experiment 1) The PL spectra of the pristine $K_3LuSi_2O_7:Eu$ and the sample treated in degradation conditions at 80% relative humidity (RH), 80 °C for different time, respectively. The inset shows the dependence of normalized integrated PL intensities on the time. c Digital photographs of these samples under natural light.

Reference Fig. 7 a The PL spectra of freshly prepared $K_3LuSi_2O_7:0.01Eu$ and the samples exposed to the air for 60 days, under 394 nm excitation. **b** Normalized time-resolved luminescence spectra of $K_3LuSi_2O_7:0.01Eu$ under 450 nm excitation.

- Above the $x = 0.01$ Eu level the emissiveness of the materials decrease. The authors suggest that this is due to concentration quenching. This statement should be supported by a calculation of the average Eu-Eu distances at the various doping levels. The dilution of the Eu(II) dopant with an appropriate spectroscopically silent analogue may be possible, although that would require a full characterization of the new materials.

Author reply: Thanks for your informative suggestions.

According to the suggestion, the average Eu-Eu distances (R_C) at the various doping levels are evaluated using the following equation:²

$$R_C = 2 \left[\frac{3V}{4\pi X_C N} \right]^{1/3}$$

where X_C is the doping concentration of activator ions; N is the number of cations which can be substituted by the dopant in per unit cell; and V is the volume of the unit cell. In the case of $K_3LuSi_2O_7:Eu$ phosphors, $N = 3$. The values of R_C are calculated to be 18.46, 14.65, 12.80, 11.63, 10.79 Å for $x = 0.01, 0.02, 0.03, 0.04$ and 0.05 samples, respectively. Thus, the concentration quenching is not triggered by the exchange interaction type due to the corresponding critical distance for exchange interaction is about 3-5 Å.

According to Dexter's theory, the type of electric multipolar interaction can be calculated by using the following formula:³

$$I/x = K[1 + \beta(x)^{\theta/3}]^{-1}$$

where x is the activator concentration, I is the emission intensity, K and β are constants for the same excitation condition for a given host lattice; $\theta = 3$ stands for energy transfer among the nearest neighbor ions, while $\theta = 6, 8$ and 10 stands for dipole-dipole, dipole-quadrupole and quadrupole-quadrupole interactions, respectively. Reference **Fig. 8** shows the dependence of $\log(I/x)$ versus $\log(x)$. The value of θ is calculated to be 3.6. It indicates that energy transfer among the nearest neighbor ions accounts for the concentration quenching mechanism in $K_3LuSi_2O_7:Eu^{2+}$ phosphors. Moreover, the previous analysis found that Eu^{3+} exist in $K_3LuSi_2O_7:Eu^{2+}$. So, the

concentration quenching may also be affected by the presence of Eu^{3+} ion. We have added these contents in the revised manuscript and supporting information.

Reference Fig. 8 The curve of $\log(I/x)$ vs. $\log(x)$ in $\text{K}_3\text{LuSi}_2\text{O}_7:\text{Eu}^{2+}$ phosphors.

- And as a follow-up to the previous questions: can the quenching at high Eu levels be due to the presence of Eu(III) contaminants? These may be difficult to see in steady-state mode, but their presence may be probed by time-resolved experiments with longer delays and direct excitation at 394 nm of a putative Eu(III) center.

Author reply: Yes, we agree with the comment of the reviewer.

The PL spectra (Reference **Fig. 7a**) under 394 nm excitation and time-resolved luminescence spectra (Reference **Fig. 7b**) discussed above verified the existence of Eu^{3+} center. The presence of the coexistence of Eu^{2+} and Eu^{3+} can be probed by time-resolved experiments as suggested by the reviewer. Moreover, the concentration quenching may be also affected by the presence of Eu^{3+} ion.

Question 8

References: Does this manuscript reference previous literature appropriately? If not, what references should be included or excluded?

The references are appropriate. The ones cited in the Introduction provide a fair overview of the field and are representative of the state of the art. In general, the authors provide good supporting references for their statements. The only exception is the discussion on the effects of coordination geometry on the Eu(II) spectroscopy (in the 'Mechanism of NIR emission...'), where the statement on the average bond length and distortion should be supported by a reference. (bottom of page 3) On a different note, a number of the cited references are missing page numbers and/or DOIs. This is presumably an EndNote problem, but should be fixed.

Author reply: Thanks for the suggestions.

We have cited the relative reference to support the statement on the average bond length and distortion (Reference 31). We have also carefully checked the format of all the references. Thanks for your careful checking.

Question 9

Clarity and context: Is the abstract clear, accessible? Are abstract, introduction and conclusions appropriate?

The abstract should be somewhat re-written, as the first half reads more like an abbreviated introduction. Specifically, references to Ce^{3+} should be removed, as it is not used in the work. Similarly, the text on the strong nephelauxetic effect and crystal field splitting, as it is too

technical.

The term 'abnormal NIR emission' should be replaced with 'unusual NIR emission', or 'unexpected NIR emission', or even better, 'unprecedented NIR emission'; as the emission is well explained in the manuscript, so is not abnormal (in fact, in line with the authors' hypothesis).

Author reply: Thanks for the suggestion.

We have re-organized the sentences in the abstract in the revised manuscript. Thanks for the correction and informative comment.

The introduction and the conclusions are appropriate in terms of general content. I have the following minor comments about the text:

- The first sentence is too long, and is too general. It should be broken up into 2-3 shorter sentences. Furthermore, the end 'and the unique NIR absorption, etc.' is too vague.
- There are a number of instances where subject and verb are unmatched, i.e. subjects in singular are paired with verbs in plural and vice versa.
- A few words that are too colloquial. E.g.: 'famous activators' (page 1), 'it is easy to understand' (page 2), 'super broad excitation' (several instances).

Author reply: We have reorganized these sentences in the revised manuscript. Thanks for the suggestions.

- Figure 2 is not referenced in the text. The references made to Figure 1 in the Photoluminescence properties section should be to Figure 2.

- On page 3, in column, the occupation mechanism is given. It is unclear what the different symbols mean in the equation. What are Eu'K, Eu*Y, etc? Are these really Y?

Author reply: We are very sorry for these mistakes. We have updated the information on Figure 1 and Figure 2.

The occupation mechanism is proposed as the synergetic effect of $\text{Lu} \rightarrow \text{Eu}$, and $\text{K} \rightarrow \text{Eu}$ replacements in $\text{K}_3\text{LuSi}_2\text{O}_7$. The symbols represent the occupation of replaced cations. In order to avoid the misleading, we deleted the equation and just described the occupation mechanism in revised manuscript.

- Please round the wavenumbers to the nearest 50 cm^{-1} .

- Please round the lifetimes as appropriate for the experimental setup.

Author reply: Thanks for your suggestions. We have revised these problems in the revised manuscript.

- In figures with spectra, please provide a scale with both nm and cm^{-1} (possibly with nm at the bottom and cm^{-1} at the top).

- In the legend of Figure S3 it should be 'reflectance' and not 'reflection'.

- In Figure 3c 'Blue shif' should be 'Blue shift'

Author reply: Thanks for the correction.

We have provided a scale with both nm and cm^{-1} in figures with spectra.

The spelling errors have been revised accordingly.

- Please provide details on how the intensity was normalized in Figure S3b. Given the noise in the

spectra, it would be helpful to provide the ratio of the 300 and 440 nm (or some wavelengths where the data are not as noisy) at the various emission wavelengths.

Author reply: In order to avoid noise from the instrument, the excitation intensity of 430 nm was used to normalize each PLE spectrum. Thanks for the suggestion.

- On page 2, in the discussion of the crystal structure: the sizes of the ions are compared, without giving the actual values. It would be important to have the ionic radii of K^+ , Eu^{2+} , Lu^{3+} and Rb^+ in the text for ready comparison.

Author reply: As suggested, we have added the ionic radii values of Eu^{2+} , K^+ , Rb^+ , Y^{3+} , Lu^{3+} in supporting information.

Reference Table 2 The ionic radii of Eu^{2+} , K^+ , Rb^+ , Y^{3+} , Lu^{3+} in the different fold of coordination.

Ion	Coordination	Ionic radii (Å)
Lu^{3+}	6	0.86
Y^{3+}	6	0.90
K^+	6	1.37
K^+	9	1.55
Rb^+	6	1.52
Rb^+	9	1.63
Eu^{2+}	6	1.17
Eu^{2+}	9	1.30

Question 10

Please indicate any particular part of the manuscript, data, or analyses that you feel is outside the scope of your expertise, or that you were unable to assess fully.

I am not a structural chemist, and am not competent to evaluate the quality of the XRD data.

Author reply: Thanks for the comment. All the XRD data are validity.

Question 11

Is the manuscript clearly written? If not, how could it be made more clear or accessible to nonspecialists?

Yes. The abstract and the introduction should be slightly rephrased e.g. as suggested above.

Author reply: Thanks. We have revised them.

Question 12

Would readers outside the discipline benefit from a schematic of the main result to accompany publication?

Possibly.

Author reply: Thanks.

Question 13

Could the manuscript be shortened? (Because of pressure on space in our printed pages we aim to

publish manuscripts as short as is consistent with a persuasive message.)

The manuscript is quite concise, and I'm reluctant to suggest substantial shortening. However, the fact that there are two distinct emitters in the materials is discussed in several places and in different contexts. It should be possible to combine these into a shorter section.

Author reply: Thanks for the positive comment. The mentioned contents have been re-organized appropriately. We just highlight the analysis of two emission centers in the structural description. Thanks again for your suggestion.

Question 14

Should the authors be asked to provide supplementary methods or data to accompany the paper online?

They should provide more data on the synthesis and the handling of the isolated materials. For example the scale of the syntheses, the number of times they were repeated, the yield of product, isolation. This is currently missing, and it makes the synthetic protocols irreproducible.

Author reply: We have repeated the synthesis for many times (above 30 times and above 100 samples) by high temperature solid-state reaction, and we can successfully obtain the samples. Moreover, different persons can also repeat the synthesis according to the given experimental conditions.

As for the scale of the syntheses, 3.8 g raw materials (K_2CO_3 , Lu_2O_3 , SiO_2 , Eu_2O_3) can produce about 2.3 g products ($K_3LuSi_2O_7:Eu$). So, the scale of the syntheses sizable can meet the needs of experiment. As shown in the following reference **Fig. 9**, we can find the as-obtained samples.

Reference Fig. 9 The photos for the as-prepared $K_3LuSi_2O_7:Eu^{2+}$ phosphors samples with the crucible.

Question 15

Have the authors done themselves justice without overselling their claims?

Yes.

Author reply: Thanks.

Question 16

Have they been fair in their treatment of previous literature?

Yes.

Author reply: Thanks.

Question 17

Have they provided sufficient methodological detail that the experiments could be reproduced?

Please see the comment above on the synthesis. I'd also prefer to see more detail on the photophysical studies, although there is quite a bit of information in the figure legends. Specifically, the variable temperature experiments should be discussed (how were these performed, how was cooling provided, etc). How was the lifetime fitting performed? Which software was used for data acquisition and processing?

Author reply: Thanks for the suggestions. The mentioned details have been responded in the corresponding "author replay" above. We have also added some revisions in the revised manuscript and SI file.

As for another detail on the luminescence thermal quenching behavior of the sample, it is measured by FLS 920 (Edinburgh Instruments) equipped with a MercuryITc temperature control instrument (OXFORD). Liquid nitrogen was used to cool temperature. Lifetime fitting is performed by the FAST software (FLS 920 attachment).

We have also added related contents in the revised manuscript,

Question 18

Is the statistical analysis of the data sound, and does it conform to the journal's guidelines?

Not generally applicable, but see comment on the standard deviation of the lifetime data.

Author reply: The standard error of lifetime have been added in the supporting information.

Question 19

Are the reagents generally available?

Yes.

Author reply: Thanks.

Question 20

Are there any special ethical concerns arising from the use of human or other animal subjects?

No.

Author reply: Thanks.

Answers to Reviewer #3 Comments:

Manuscript under the title “Eu²⁺-Doped Near-Infrared-Emitting Phosphor for Light-Emitting Diodes” reports synthesis and optical properties of the target materials. From the application point of view, the material is very interesting. However, there are some major issues in the manuscript that authors should take care of.

Question 1

The English should be carefully re-inspected, since there are many grammar and punctuation mistakes. Besides, some figures in the manuscript are assigned incorrectly.

Author reply: We have checked and revised grammar and punctuation in the revised manuscript. Figures in the manuscript are also checked. We believe that this revised version can be acceptable. Thanks for the suggestions.

Question 2

Authors state that Eu²⁺ ions occupies two different sites, namely, K₂O₆ and LuO₆. However, Rietveld refinement data presented in Table S2 show, that the refinement was performed only as Eu²⁺ ions were occupying Lu³⁺ sites. This contradicts the discussion in the manuscript.

Author reply: Thanks for your informative comments. The similar question has been also proposed by the Reviewer #1, and some detailed analysis can be also found “author reply” there. As mentioned above, it is indeed difficult to determine the exact occupancy of activators by XRD refinement when the doping concentration is very low, especially when there exist multiple luminescent centers. There are three cationic sites (K1, K2, Lu) in the studied K₃LuSi₂O₇ host, which may be occupied by Eu ions. So, the accurate occupancy of Eu is difficult by XRD refinement. But we can make sure that Eu ions occupy the small Lu ions in K₃LuSi₂O₇, because the cell volumes increased with the doping concentration, as given in the above Reference **Fig. 1a**. So, we only considered the sites occupation of Eu ions in Lu sites previously. However, we can infer that Eu occupy K2 and Lu sites from the combined results on the change of unit cell volumes (Reference **Fig. 1a**) and bond lengths (Reference **Fig. 1b**). Firstly, if Eu²⁺ only occupy small Lu sites, the cell volume will show a linear increasing trend with the increase of Eu doping concentration. However, the nonlinear increasing in Reference **Fig. 1a** indicates that Eu also occupy the large K ions. Secondly, the bond length of K2-O and Lu-O changed with the doping concentration Reference **Fig. 1b**, which further certify that Eu occupy K2 and Lu sites in K₃LuSi₂O₇. These contents have been discussed clearly in manuscript. Anyway, to avoid misleading readers, we have performed the Rietveld refinement again and deleted the sites occupancy of Eu ions in Lu sites, as shown in **Table S2**.

Question 3

My major concern is with the spectroscopic data of the synthesized materials:

a) Fig. 2a shows the excitation spectrum of $\text{K}_3\text{LuSi}_2\text{O}_7:0.01\%\text{Eu}^{2+}$ compound. However, there are some sharp lines ca. 450 nm. Where these lines come from? Eu^{2+} ions possess no sharp lines in excitation spectra, thus it should be instrument/measurement related. Were excitation spectra corrected for instrument response?

Author reply: Yes, the sharp lines at 450 nm come from the instrument used. Moreover, both excitation and emission spectra were corrected for instrument response.

b) The reflection spectra given in Fig. S3a are also rather unusual, especially the one for undoped sample. It is strange, that the reflectance of undoped sample is the highest at ca. 250 nm and then decreases. More commonly, the results are opposite, i.e., the reflectance at short wavelengths for undoped materials is lower if compared to longer wavelengths. Besides, what is the bandgap of undoped material? The reflectance values of the synthesized samples at longer wavelengths is ca. 90%. This usually suggests that some defects are found in the material. What was the body colour of the synthesized materials? Was it greyish?

Author reply: Thanks for the informative suggestions. Normally, the reflectance at short wavelengths for host materials is lower compared to longer wavelengths. We re-measured reflection spectra of $\text{K}_3\text{LuSi}_2\text{O}_7$ host and BaSO_4 , as shown in following reference **Fig. 10**. The reflectance of $\text{K}_3\text{LuSi}_2\text{O}_7$ increase slightly at 245 nm and then decrease along the short wavelength. As for the reflectance values at longer wavelength is about 80%-90%. It is acceptable.

On one hand, the reflectance values of BaSO_4 is defined as 100%. On the other hand, the reflection spectra of $\text{K}_3\text{LuSi}_2\text{O}_7$ host was measured with the BaSO_4 as base line. So, the reflectance values of host materials at longer wavelength is lower than 100%. The body color is white under natural light. The band gap is estimated by the following equation:⁴

$$\frac{[F(R_\infty)h\nu]^2}{A} = h\nu - E_g.$$

The value of the band gap E_g is about 5 eV.

Reference Fig. 10 Reflectance spectra of BaSO_4 and $\text{K}_3\text{LuSi}_2\text{O}_7$ host. The inset shows the photograph of $\text{K}_3\text{LuSi}_2\text{O}_7$ host.

c) Fig. 2c shows the emission spectra fit with two Gaussian curves. Since the wavelength was converted to energy scale, were the intensity also corrected for such conversion? (see, for instance, 10.1016/j.optmat.2012.01.034).

Author reply: Yes, the intensity was also corrected for such a conversion from wavelength to

energy scale. The intensity has also be corrected. The two broad fitting curves are due to the broad emission band from 550 to 850 nm, as discussed in this manuscript.

d) The PL lifetime values of Eu^{2+} doped phosphors in the red spectral region usually are around 1 μs . However, the authors report significantly lower values. What could be the reason for such mismatch?

Author reply: Thanks very much for your informative suggestions.

Yes, the PL lifetime values of Eu^{2+} doped phosphors in the red spectral region usually are around 1 μs . The relationship of decay times τ and emission wavelength λ can be described by following equation.^{5,6}

$$\Gamma = \frac{1}{\tau} \propto \frac{n}{\lambda^3} \left(\frac{n^2+2}{3} \right)^2 | < 5d|\mu|4f >|^2.$$

This equation predicts a longer decay time with longer emission wavelength. Thus, for the near infrared $\text{K}_3\text{LuSi}_2\text{O}_7:\text{Eu}$ phosphor, the decay times should be longer than 1 μs .

As proposed in the Question 4 by Reviewer #2 and the response in the “author reply” there, our previous lifetime fitting is based on the test time (4 minutes). However, this time is too short to get enough photons to obtain accurate lifetime values, especially for the samples with weak emission intensity. During the revision, we measured the lifetime values based on the statistical photons of 5000 or 10000. It is found that the decay time value measured at 300 K is 1.58 μs , and the value is 2.28 μs at 80 K. These values should be acceptable. We have also highlighted this experimental details in the experimental section to avoid the misleading results.

We have updated these results and description in the revised manuscript.

e) Authors mentioned that two detectors were employed in PL spectra measurements, i.e., VIS and NIR. The excitation wavelength and temperature dependent emission spectra were obviously recorded using VIS detector. The sensitivity of VIS detector in the red/infrared part of the spectrum is rather low and thus the correction for instrument response (I assume authors have done that since Edinburgh Instruments spectrometers always come with emission correction files) will significantly increase the intensity in this area especially if the intensity of the sample is relatively low in this area as well. Besides, the spectra measured with this detector in not complete. Taking into account that temperature dependent emission integrals were also analyzed in Fig. 3b the significant part of emission data are not present in this figure. Therefore, the temperature dependent and excitation wavelength dependent emission should also be measured with NIR detector.

Author reply: Thanks very much for your informative suggestions.

Yes, to obtain the full spectra measured with NIR detector is important. In our manuscript, PL and PLE spectra were corrected by the correction files attached with FLS 920 instruments. The emission spectra of the fabricated NIR pc-LED was measured by a fiber spectrophotometer (NOVA high sensitive spectrometer, idea optics, China), and 450 nm laser diode was used as the excitation source. Unfortunately, we only have two different laser diodes of 375 nm and 450 nm. So, only the emission spectra that excited at 375 nm and 450 nm were measured by fiber spectrophotometer, as shown in the following reference **Fig. 11**. The spectral profiles are similar with the emission spectra measured at FLS 920, and the tiny difference is due to the response variation for different instrument. The relative contents have been added in the revised manuscript

and supporting information.

Moreover, the fiber spectrophotometer are not equipped with temperature control equipment. Therefore, the temperature dependent emission spectra were not measured. As for the thermal stability, although the spectra after 850 nm is missing, the existing temperature-dependent spectra are enough to evaluate its thermal stability. We do believe the present data is suitable for the results and hope the reviewer can accept this. Thanks.

Reference Fig. 11 Normalized PL spectra of $K_3LuSi_2O_7:Eu$ measured with fiber spectrophotometer under different excitation wavelength, 375 nm and 450 nm.

f) Authors also discuss the activation energy of thermal quenching induced by photoionization referring to model proposed by P. Dorenbos. However, even though the temperature dependent integrated emission data are given, the activation energy was not calculated. Why?

Author reply: Thanks for your suggestions.

According to Dorenbos's model, the activation energy ΔE was derived from the following empirical formula:⁷

$$\Delta E = \frac{T_{0.5}}{680} eV$$

where $T_{0.5}$ is the quenching temperature, at which the emission intensity drops to 50% of its original intensity. The value of ΔE is calculated about 0.66 eV.

We have added this content in the revised manuscript.

Reference

1. Ding, X., Wang, Y. Commendable Eu^{2+} -Doped Oxide-Matrix-Based $LiBa_{12}(BO_3)_7F_4$ Red Broad Emission Phosphor Excited by NUV Light: Electronic and Crystal Structures, Luminescence Properties. *ACS Appl. Mater. Interfaces* **9**, 23983-23994 (2017).
2. Blasse, G. Energy transfer in oxionic phosphors. *Philips Res. Rep.* **24**, 131-144 (1969).
3. Dexter, D. L. A Theory of Sensitized Luminescence in Solids. *J. Chem. Phys.* **21**, 836-850 (1953).
4. Tauc, J., Radu Grigorovici, Anina Vancu. Optical properties and electronic structure of amorphous germanium. *Phys. Stat. Sol.* **15**, 627-637 (1966).
5. Duan, C.-K., Reid, M. F. Local field effects on the radiative lifetimes of Ce^{3+} in different hosts. *Current Appl. Phys.* **6**, 348-350 (2006).
6. Dorenbos, P. Fundamental Limitations in the Performance of Ce^{3+} -, Pr^{3+} -, and Eu^{2+} -Activated Scintillators. *IEEE Trans. Nucl. Sci.* **57**, 1162-1167 (2010).

7. Dorenbos, P. Thermal quenching of Eu^{2+} 5d–4f luminescence in inorganic compounds. *J. Phys.: Condensed Matter* **17**, 8103 (2005).

REVIEWERS' COMMENTS:

Reviewer #2 (Remarks to the Author):

The authors have seriously engaged with my comments. They have performed additional experiments that support their original hypothesis. New data are included in the manuscript as well as the supporting information. They have revised the text, which now reads much more smoothly. I'm supportive of their manuscript being published in Nature Communications.

I have one small comment:

In the supporting info, in Table S5 the lifetimes and the errors are not given to the same number of significant figures (the error is more precise). This should be fixed.

Reviewer #3 (Remarks to the Author):

The authors have appropriately addressed the issues raised before. In my opinion the manuscript was significantly improved. The results are novel, properly discussed and the conclusions are well supported by the experiments.

My only suggestion is to merge equations (1), (2), (3) into one. These expressions are very basic and well known, therefore, there is no need to repeat them three times.

Answers to Reviewer #2 Comments:

The authors have seriously engaged with my comments. They have performed additional experiments that support their original hypothesis. New data are included in the manuscript as well as the supporting information. They have revised the text, which now reads much more smoothly. I'm supportive of their manuscript being published in Nature Communications.

I have one small comment:

In the supporting info, in Table S5 the lifetimes and the errors are not given to the same number of significant figures (the error is more precise). This should be fixed.

Author reply: We sincerely thank the reviewer for the positive evaluation of our manuscript. We have fixed the lifetimes and the errors in the same number of significant figures (Table S5).

Answers to Reviewer #3 Comments:

The authors have appropriately addressed the issues raised before. In my opinion the manuscript was significantly improved. The results are novel, properly discussed and the conclusions are well supported by the experiments.

My only suggestion is to merge equations (1), (2), (3) into one. These expressions are very basic and well known, therefore, there is no need to repeat them three times.

Author reply: Thanks, the equations (1), (2), (3) have been merged into one, as shown follow, and the corresponding descriptions have been also added.

$$I = \sum_{i=1}^n A_i \exp\left(-\frac{t}{\tau_i}\right), \quad (n = 1, 2, 3)$$